# A Novel Interactive Fusion Method with Images and Point Clouds for 3D Object Detection

**Kai Xu [1,2], Zhile Yang [1], Yangjie Xu [1] and Liangbing Feng [1,*]**

[1]   Shenzhen Institute of Advanced Technology, Chinese Academy of Sciences, Shenzhen 518055, China;
     xk1992@mail.ustc.edu.cn (K.X.); zl.yang@siat.ac.cn (Z.Y.); yj.xu@siat.ac.cn (Y.X.)

[2]   School of Software Engineering, University of Science and Technology of China, 188 Renai Road,
     Suzhou 215123, China

*   Correspondence: lb.feng@siat.ac.cn

**Abstract:** This paper aims at tackling the task of fusion feature from images and their corresponding point clouds for 3D object detection in autonomous driving scenarios based on AVOD, an Aggregate View Object Detection network. The proposed fusion algorithms fuse features targeted from Bird's Eye View (BEV) LIDAR point clouds and their corresponding RGB images. Differing in existing fusion methods, which are simply the adoption of the concatenation module, the element-wise sum module or the element-wise mean module, our proposed fusion algorithms enhance the interaction between BEV feature maps and their corresponding image feature maps by designing a novel structure, where single level feature maps and utilize multilevel feature maps. Experiments show that our proposed fusion algorithm produces better results on 3D mAP and AHS with less speed loss compared to the existing fusion method used on the KITTI 3D object detection benchmark.

**Keywords:** fusion; point clouds; images; object detection

## 1. Introduction

It is a fact that deep neural networks rely on a large number of data to guarantee training effectiveness [1]. In general, the more data that is fed, the better the performance that will be obtained, particularly when feeding abundant sensor data to the network model. In the field of self-driving cars or 3D object detection, the camera and lidar are dominant sensors. RGB images from cameras contain rich texture information of the ambience, whereas the depth is lost. Point clouds from lidar can provide accurate depth and reflection intensity descriptions, but the resolution is comparatively low. Naturally, the effective fusion [2] of these sensors would be expected to deal with the drawbacks of a single sensor in complicated driving scenarios.

There are three major fusion algorithms to solve multi-sensor fusion problem including early fusion [3–7]; late fusion [8–10] and deep fusion [11]. In the early fusion architecture, first, features from single sensor concatenate or element-wise sum (mean) the features from other sensors; second, the outputs of fused feature maps would be sent to classification or segmentation. An advantage of early fusion is that the joint feature space between the modalities is potentially more expressive. However, the learning problem becomes more difficult due to the fact that the classifier must learn a mapping from a higher-dimensional feature space. Late fusion usually has a multi-net-branch and each network branch is first run on its network structure in the corresponding sensing modality separately, then feature maps from each branch would be fused by concatenation or element-wise sum (mean) as final input to classification or segmentation. Compared to early fusion, late fusion is easier to learn, but less expressive and sometimes the former could utilize more data than the latter one. Especially when training data is not sufficient, the late fusion performs more effectively. The recent deep fusion

inspired by refs. [12,13] uses an element-wise mean for the join operation. Besides, the three branches, using fused feature as unify input, would be trained dependently then combined with the element-wise mean and iteration. Deep fusion makes features from different views interact frequently, but in each interaction, there is also a linear model which is the same as early fusion and late fusion. The linear model is flexible with simple accomplishment, however, it is much less expressive than the nonlinear model, which may suffer from more time and memory cost.

Our proposed fusion algorithm aims at combining the linear model and the nonlinear model, and enhancing the interactions between image features and their corresponding point clouds features, while the independence of multi-view features is kept at the same time. The proposed fusion algorithm is elaborated in the following section.

## 2. Related Work and Proposed Method

There are a few works that exploit multi-sensor data including the combination of RGB images and depth images and the fusion of RGB images and point clouds [1]. Ref. [7] utilizes RGB images and depth images with early fusion strategy and trains pose-based classifiers for 2D detection [14]. Refs. [4,5] applying early fusion strategy by projecting the point cloud to the plane and augmenting the image channels after upsampling. Ref. [10] fuses images and point clouds by late fusion strategy for urban segmentation and computes size, shape, position, color features, a high-dimension Bag-of-words (BoW) descriptor in images and point clouds. Ref. [11] fuses lidar bird view, lidar front view and image for 3D object detection with deep fusion. Besides, it projects point cloud to the bird's eye view instead of the image plane and produces good results in KITTI [15]; however, it performs poorly in the test of average heading similarity (AHS).

We evaluated our fusion algorithm in KITTI by 3D object detection with images and point clouds based on AVOD [16], a two-stage 3D object detector for autonomous driving scenarios on 3D localization, orientation estimation, and category classification tasks with a low memory overhead and a linear fusion algorithm. Different from existing fusion algorithms, our fusion algorithms combine a linear model and nonlinear model and enhance interaction and independence. We can find a detailed visual structure for one of our proposal fusion algorithms in Figure 1. It could be observed that two types or one group input feature maps to fusion output feature maps, our proposed fusion algorithms contain a linear model element-wise mean, and concatenation and nonlinear model convolution with a Relu [17] activation function. Besides, it reinforces interaction and independence by adding one type feature maps into the other type and considering the mixing ratio, then the two types of fused feature maps are separately through convolution layers. Finally, the two branches fuse again by an element-wise mean. We name this framework as the single level feature maps fusion algorithm. In addition, the multilevel feature maps fusion algorithm is illustrated in Figure 2, where four type or two group input feature maps are adopted for fusion of output feature maps. The proposed multilevel fusion algorithm contains la inear model element-wise mean, as well as concatenation and a nonlinear model convolution with a Relu [17] activation function. Moreover, it reinforces interaction and independence by adding one type feature maps into the other type and considering the mixing ratio, then the four type fused feature maps are inputted through convolution layers separately, finally, the four branches fuse again by the element-wise mean.

We examined our proposed fusion algorithm by replacing the two fusion parts of AVOD [16] with our fusion algorithm. To be specific, the single level feature maps fusion algorithm uses the Figure 1 structure to displace the second fusion part of AVOD [16], and the multilevel feature maps fusion algorithm uses the Figure 2 structure to displace the first fusion part of AVOD.

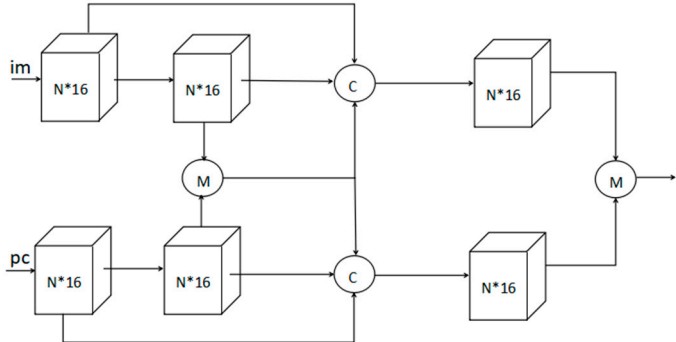

**Figure 1.** Visual structure of the fusion algorithm with single level feature maps: first, image feature maps and point cloud feature maps fuse by element-wise mean. Then, the fused part concatenates with the preceding two layer feature maps. Next, each of the two branches is fed through a convolution layer to reorganize the feature maps, finally the two branches fuse again by element-wise mean to the region proposal.

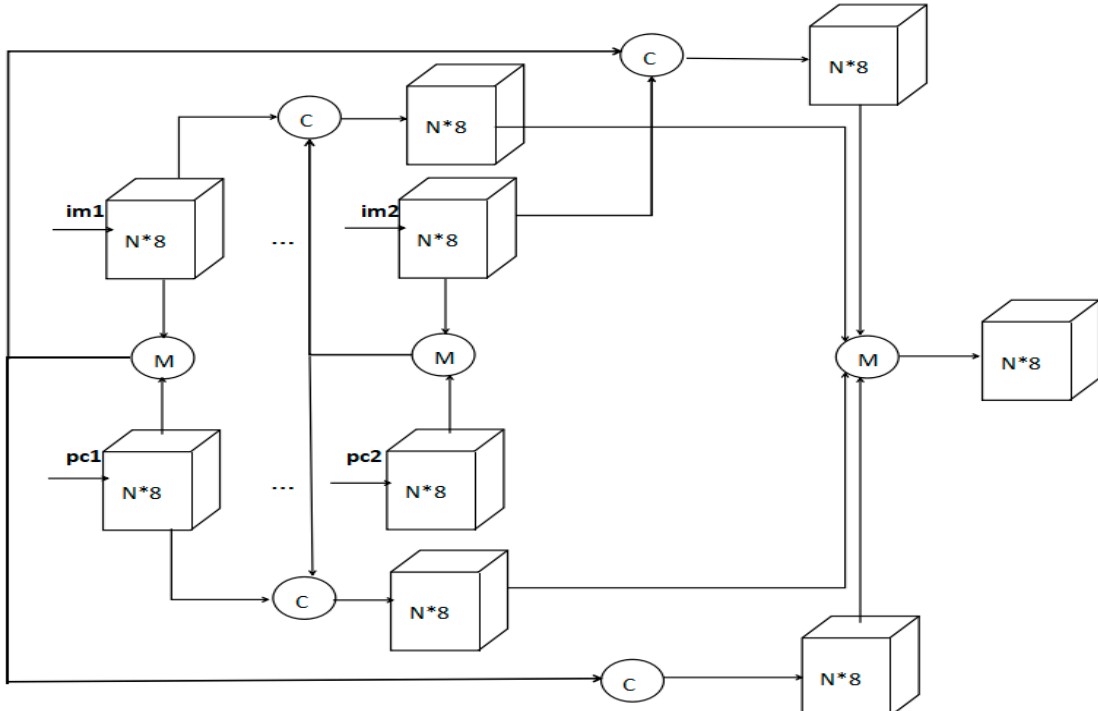

**Figure 2.** Visual structure of the fusion algorithm with multilevel feature maps: first, image feature maps and point cloud feature maps in each group are fused by element-wise mean separately (im1 and pc1 are of one group, and im2 and pc2 are of another group). Then, the fused part concatenates with another corresponding level feature maps. Next, each of the four branches is fed through the convolution layer to reorganize the feature maps. Finally, the four branches fuse again by element-wise mean to the region proposal.

## 3. Experiments

We elevated our fusion algorithm on published AVOD [16] where the metrics are 3D AP (average precision) and AHS (average heading similarity). The descriptions of the metrics are in the formulas below,

$$AP = \tfrac{1}{11} \sum_{r \in \{0,0.1,\dots,1\}} \max_{\hat{r} : \hat{r} \geq r} P(\hat{r})$$

$$AHP = \tfrac{1}{11} \sum_{r \in \{0,0.1,\dots,1\}} \max_{\hat{r} : \hat{r} \geq r} S(\hat{r})$$

In the two formulas, $r$ means recall, $P(r)$ means precision at the recall equal to $r$ and $S(r)$ means orientation similarity at the recall equal to orientation similarity. To get the AP we calculate the IoU between the ground truth and test results in 3D space. Because of not considering the direction in AP, we use AHP. It is used to measure the similarity between the test results and the ground truth in terms of direction.

To test our proposed fusion algorithms, we replace AVOD [16] data fusion parts by our proposed fusion algorithms and use the default hyper-parameter, training set and validation set, the same with AVOD, to eliminate other influences and the performances on KITTI val set are shown in Table 1, Table 2, Figures 3 and 4.

**Table 1.** Our proposed single level fusion algorithm evaluation on the car class in the validation set. For evaluation, we show the AP and AHS (in %) at 0.7 3D IoU.

|  | Easy | | Moderate | | Hard | |
| --- | --- | --- | --- | --- | --- | --- |
|  | AP | AHS | AP | AHS | AP | AHS |
| MV3D [11] | 83.87 | 52.74 | 72.35 | 43.75 | 64.56 | 39.86 |
| AVOD [16] | 83.08 | 82.96 | 73.62 | 73.37 | 67.55 | 67.24 |
| ours | **84.16** | **84.05** | **74.45** | **74.13** | **67.80** | **67.40** |

**Table 2.** Our proposed multilevel fusion algorithm evaluation on the car class in the validation set. For evaluation, we show the AP and AHS (in %) at 0.7 3D IoU.

|  | Easy | | Moderate | | Hard | |
| --- | --- | --- | --- | --- | --- | --- |
|  | AP | AHS | AP | AHS | AP | AHS |
| MV3D [11] | 83.87 | 52.74 | 72.35 | 43.75 | 64.56 | 39.86 |
| AVOD [16] | 83.08 | 82.96 | 73.62 | 73.37 | 67.55 | 67.24 |
| ours | **84.62** | **84.41** | **74.88** | **74.45** | **68.30** | **67.79** |

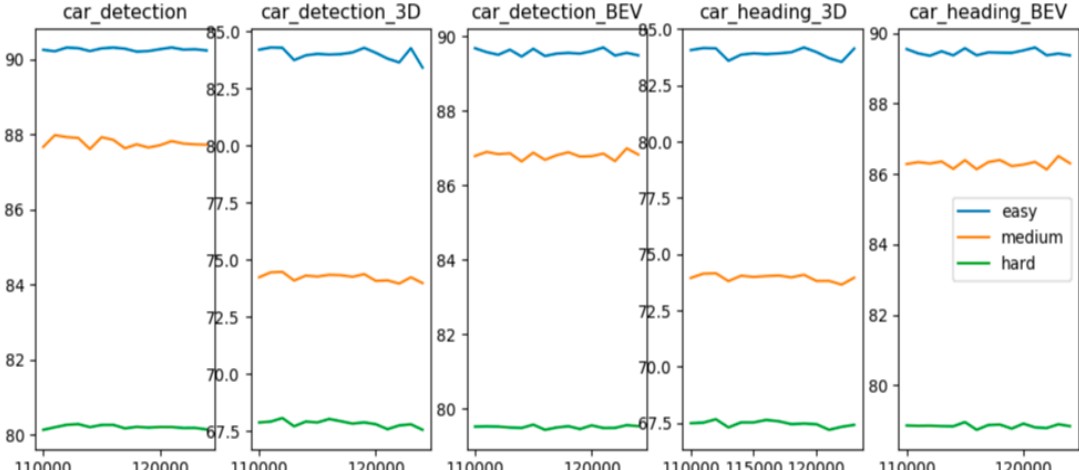

**Figure 3.** Multilevel fusion algorithm AP vs. Step, and print of the 5 highest performing checkpoints for each evaluation metric.

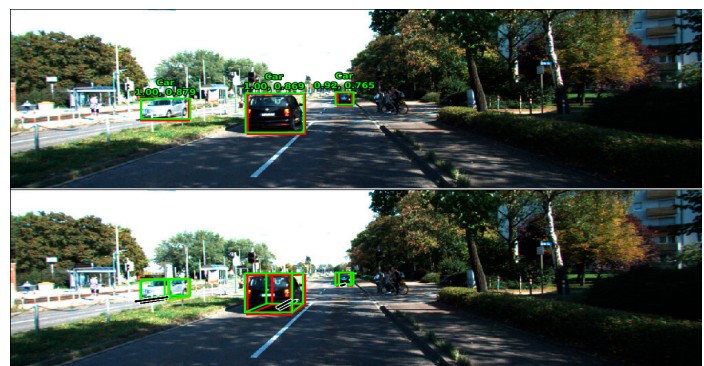

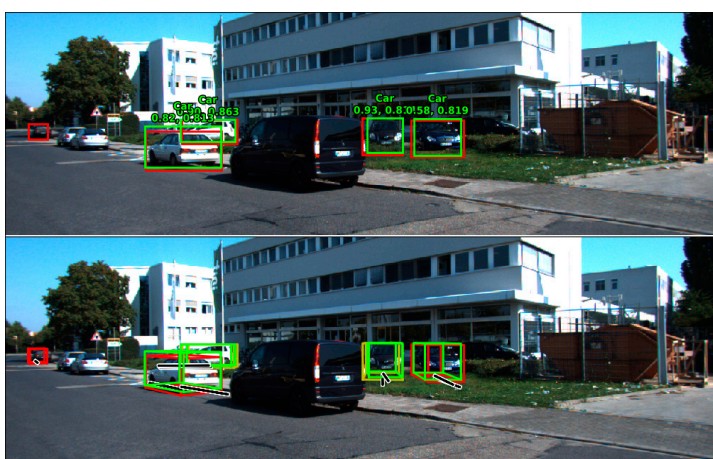

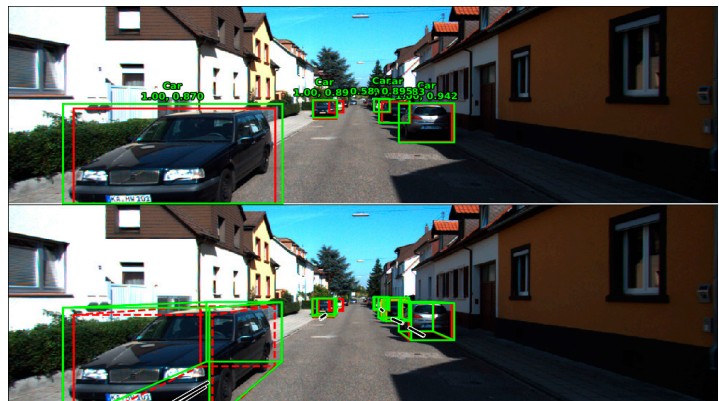

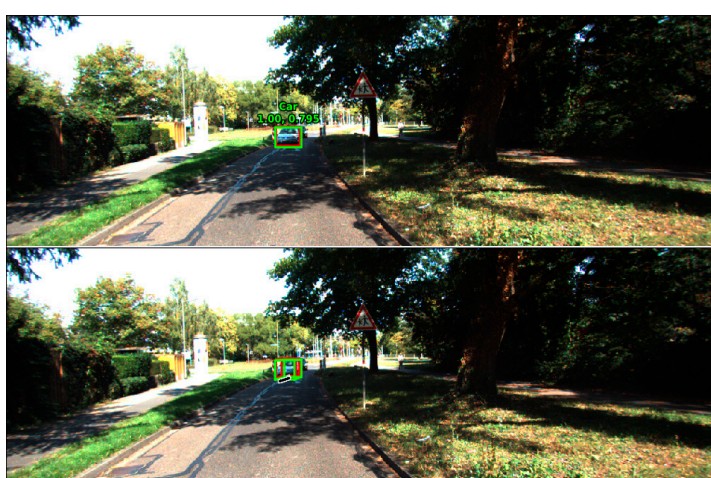

**Figure 4.** *Cont.*

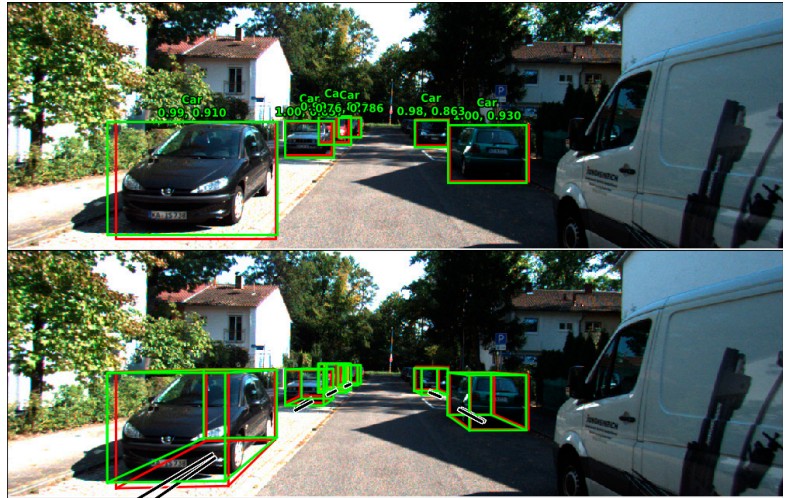

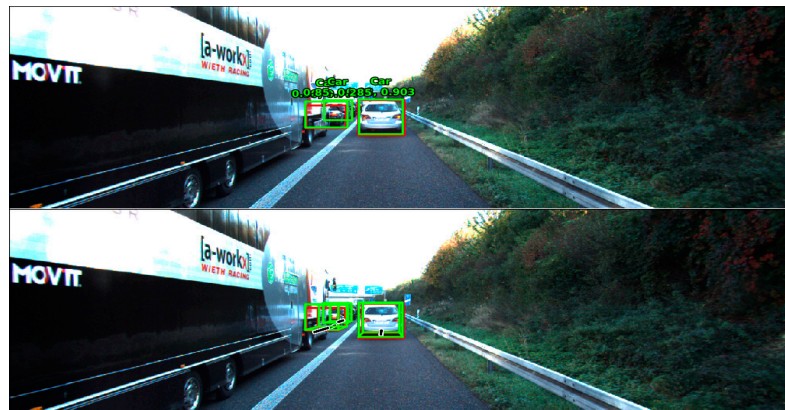

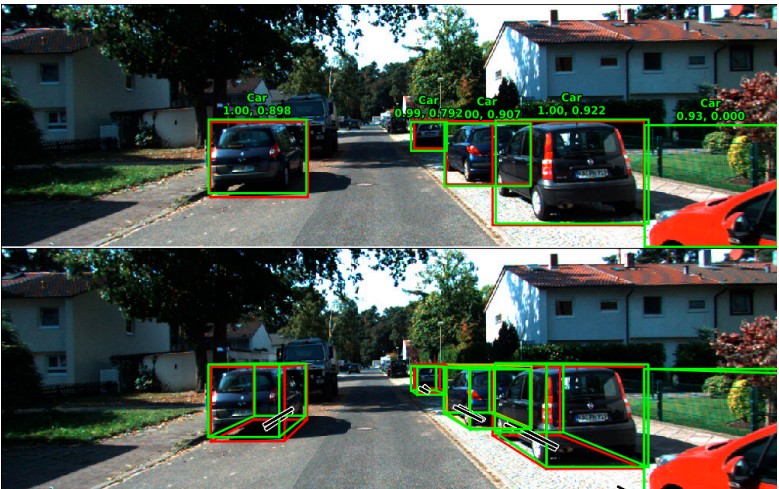

**Figure 4.** Visualization of multilevel fusion algorithm results on KITTI val set, including 2D localization, category classification, 3D localization, orientation estimation, and category classification.

*3.1. Kernel Design*

Our proposed single level fusion algorithm, which is depicted by Figure 1, uses uniform convolution layers that are designed with kernel size 1*1, stride size 1, the output feature map size is 16, and its input feature map comes from the last feature extraction layer whose feature map resolution is the same as the original network input. The multilevel fusion algorithm, which is depicted by Figure 2, uses uniform convolution layers that are designed with kernel size 1*1, stride size 1, output feature

map size 8, and its input feature maps come from the last feature extraction layer in which the feature map resolution is the same as the original network input and the last but one feature extraction layer of which the feature map resolution is only half of the original network input.

In AVOD [16], the feature map size cropped for the proposal region is 3*3 and the feature map size cropped for the final prediction is 7*7. During training, we investigated how the kernel with different sizes affects the performance and how the kernel group inspired by GoogLeNet [18] performs. The baseline 1 diverse kernel size multilevel fusion algorithm has 1*1, 3*3 kernel in the first fusion part and 1*1, 3*3, 5*5, and 7*7 kernel in the second fusion part. The baseline 2 diverse kernel size multilevel fusion algorithm has 1*1, 3*3 kernel in the first fusion part and 1*1, 7*7 kernel in the second fusion part. The outputs of the proposal are the same as the outputs of AVOD. Our regression targets are therefore ($\Delta$x 1 ... $\Delta$x 4, $\Delta$y 1 ... $\Delta$y 4, $\Delta$h 1, $\Delta$h 2), the corner and the height offsets from the ground plane between the proposals and the ground truth boxes. The results are show in Table 3. It can be seen that our approach obtained the best result in the ratio of feature maps size and zero padding. The higher ratio, means that the number of zero by zero padding is equal or greater than the number of elements in the feature maps, and more disturbance to the feature maps caused by zero is predominant so that the original distribution of feature maps is deviated to a large extent.

**Table 3.** Our proposed multilevel fusion algorithm evaluated on the car class in the validation set compared with the baseline.

| | Easy | | Moderate | | Hard | |
|---|---|---|---|---|---|---|
| | **AP** | **AHS** | **AP** | **AHS** | **AP** | **AHS** |
| baseline1 | 83.02 | 82.84 | 73.71 | 73.13 | 67.79 | 67.15 |
| baseline2 | 84.02 | 83.84 | 74.42 | 74.03 | 68.16 | 67.74 |
| ours | **84.62** | **84.41** | **74.88** | **74.45** | **68.30** | **67.79** |

### 3.2. No BatchNorm

Batch normalization aims to eliminate the covariate shift in its input data which can improve the speed of learning and independence of each individual layer. However, we found that batch normalization regresses the 3D bounding box estimation performance. Therefore, our proposed fusion algorithms have no batch normalization layers.

### 3.3. Trainning Details and Computational Cost

To measure our proposed methods with AVOD, we just utilized the AVOD's hyperparams, like an ADAM optimizer with an initial learning rate of 0.0001 that is decayed exponentially for every 30 K iterations with a decay factor of 0.8.

Our proposed methods are based on AVOD, and, at first, we only wanted to measure the different fusion algorithm with AP, then with the improvement of AP, we reduced the runtime to 0.12 s by decreasing the output of the convolution layers in our proposed method to get a balance between speed and accuracy.

### 3.4. Architecture Design Analysis

Our fusion algorithms aim to reinforce interaction and independence of different types of feature maps and utilize linear models and nonlinear models to enhance expression. The single level fusion algorithm can approximate a general function:

$$f_{im}^{[L+1]}\left(f_{im}^{[L-1]} \otimes f_{im}^{[L]} \otimes \left(f_{im}^{[L]} \otimes f_{pc}^{[L]}\right)\right) \otimes$$
$$f_{pc}^{[L+1]}\left(f_{pc}^{[L-1]} \otimes f_{pc}^{[L]} \otimes \left(f_{pc}^{[L]} \otimes f_{im}^{[L]}\right)\right)$$

where $f_{im}^{[L+1]}$ is the image feature maps of $(L + 1)$th layer; $f_{pc}^{[L+1]}$ denotes the point cloud feature maps of $(L + 1)$th layer; $\otimes$ means concatenate or element-wise mean, and the multilevel fusion algorithm can also approximate a general function:

$$f_1^{[L+1]}\left(\left(f_{im1}^{[L-K]} \otimes f_{pc1}^{[L-K]}\right) \otimes f_{im2}^{[L]}\right)$$
$$\otimes f_2^{[L+1]}\left(\left(f_{im1}^{[L-K]} \otimes f_{pc1}^{[L-K]}\right) \otimes f_{pc2}^{[L]}\right)$$
$$\otimes f_3^{[L+1]}\left(\left(f_{im2}^{[L]} \otimes f_{pc2}^{[L]}\right) \otimes f_{im1}^{[L-K]}\right)$$
$$\otimes f_4^{[L+1]}\left(\left(f_{im2}^{[L]} \otimes f_{pc2}^{[L]}\right) \otimes f_{pc1}^{[L-K]}\right)$$

where $f^{[L+1]}$ is the feature map of $(L + 1)$th layer and the subscript means a different source of feature maps; $\otimes$ means concatenate or element-wise mean; K is an integer and less than *L*.

## 4. Conclusions

In this paper, we proposed two fusion algorithms. One is the single level feature maps fusion algorithm, and the other is the multilevel feature maps fusion algorithm. Both of the two fusion algorithms enhance interaction and independence between BEV feature maps and their corresponding image feature maps by designing a novel structure differentiated from the existing fusion methods. Our proposed fusion algorithms define a nonlinear framework to improve potential expression. The nonlinear frameworks also take advantage of linear models, being similar to the existing fusion method, for flexible interaction and reduced cost of learning. Besides, the nonlinear frameworks can be easily embedded in the CNN network so that it is frequently utilized. Experiments on the KITTI dataset show the effectiveness of our nonlinear fusion algorithms compared with the existing fusion.

**Author Contributions:** K.X. and L.F. propose algorithms and write the paper. Z.Y. and Y.X. check the algorithms and do assistant experiments.

**Funding:** This work was supported in part by the Shenzhen Science and Technology Program under Grant No. JCYJ20170811160212033, Natural Science Foundation of Guangdong Province under grants 2018A030310671 and China Post-doctoral Science Foundation (2018M631005).

**Conflicts of Interest:** The authors declare no conflict of interest.

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
