# Peer review of "A Novel Interactive Fusion Method with Images and Point Clouds for 3D Object Detection"

_applsci, doi:10.3390/app9061065_

Reviewer 1 Report

In this paper,  two different fusion algorithms for fusing pointclouds and images are introduced. I find this topic fascinating and, due to the emergence of the autonomous driving, I think it is of insterest for the scientific community. Nonetheless, the paper is hard to follow and entangled.

In fact, insufficient background and introduction is provided:
- As the proposal is a modification of AVOD, a desciption of it would be neccessary
- What are the outputs of the proposal?

In addition, as far as I know, the pointclouds are unorganized data structures, so applying convolutions to it sounds a hard goal to achive. How are organized the pointclouds so they can be forwarded to a convolution layer??

I miss more details about the KITTI dataset in the paper. Did the authors used it as provided?
Are the Easy, Moderate and Hard categories provided by the dataset? Or did the authors set it for convenience?

Furthermore, the paper also lacks of more information about the training stage: training hyperparams, optimizer of choice, epochs, training and testing loss, training and testing accuracy...

Regarding the reported results, a definition of the AP and AHS metrics must be added

The authors compared their approach to AVOD and MV3D. These methods must be described also. In addition, Table 3 shows different results for two baseline entries. What is that baseline? Why are there two different entries??

I feel that this approach could be used in an actual self driving car, so reporting the run time would help to make this paper stronger.

Finally, An extensive editing of English language and style is required.

In the light of these major concerns I cannot recommend the wwork for publication as is. However, I encourage the authors to resubmit a revised version of the paper.

Author Response

Response to Reviewer 1 Comments

Point1: As the proposal is a modification of AVOD, a desciption of it would be neccessary, What are the outputs of the proposal?

Response 1: A desciption of AVOD would be added in section2.

Point2:What are the outputs of the proposal?

Response 2: The outputs of the proposal is the same as the outputs of AVOD. Encoding 3D Bounding Box with a 10 dimensional vector((∆x 1 ...∆x 4 ,∆y 1 ...∆y 4 ,∆h 1 ,∆h 2 )).

Point3:How are organized the pointclouds so they can be forwarded to a convolution layer?

Response 3: The pointclouds are projected into image plane, named BEV(Bird’s Eye View).

Point4:about the KITTI dataset in the paper?

Response 4: In this paper, we use the open dataset in AVOD, and the link: https://github.com/kujason/avod.

Point5:information about the training stage: training hyperparams, optimizer of choice, epochs, training and testing loss, training and testing accuracy...?

Response 5: to measure our proposed methods with AVOD, we just utilize AVOD’s hyperparams, like an ADAM optimizer with an initial learning rate of 0.0001 that is decayed exponentially every 30K iterations with a decay factor of 0.8.

Point6:a definition of the AP and AHS metrics must be added?

Response 6: average precision (AP) and Average Heading Similarity(AHS).

Point7:What is that baseline? Why are there two different entries?

Response 7: the baseline is our supplementary experiment to test proposed methods, besides, baseline1 and baseline2 have different network architecture which discusses in section3.1.

Point8:the run time? 

Response 8: Our proposed methods are based on AVOD, and, at first, we just want to measure different fusion algorithm with AP, then with the improvement of AP, we reduce the runtime to 0.12s by decreasing the output of convolution layers in our proposed methods to get a balance between speed and accuracy.

Reviewer 2 Report

This paper proposes  fusion algorithms  combine the linear model and the nonlinear model. The methods  improve the interactions between image features and its corresponding point clouds features. Moreover  the independence of multi-view features is kept at the same time.

The nonlinear framework has some advantages over the linear one and it can be embedded CNN network to make it utilised frequently. The experiments appears to be good.

Here are some comments.

The computational cost and the complexity the proposed methods should be further investigated. 

The optimal kernel size should be provided. 

Author Response

Point1: The computational cost and the complexity the proposed methods should be further investigated.

Response 1: Our proposed methods are based on AVOD, and, at first, we just want to measure different fusion algorithm with AP, then with the improvement of AP, we reduce the runtime to 0.12s by decreasing the output of convolution layers in our proposed methods to get a balance between speed and accuracy.

Point2: The optimal kernel size should be provided. 

Response 2: Section3.1 had provided the optimal kernel size. In our algorithm, convolution layers that are designed with kernel size 1*1, stride size 1.

Round  2

Reviewer 1 Report

The authors addressed the concerns I raised in my previous review.
Nonetheless, I have to insist in some of my previous points:
- The AHS and AP should be clearly defined. Not in terms of semantics, but the authors should state the formulas fot both metrics.

- The 10 dimensional vector output should be clearly defined. As far as I know, a 3D bounding box would be defined by 8 points correspoding to the corners of a cube. According to this definition, I'm missing 2 dimensions.

In addition, the authors did not clearly highlighted the changes in the manuscript so I don't really know if they addressed the review properly. According to the answers, I'm assuming they did it.

I am willing to recommend for publication after the authors address these minor issues.

Author Response

Dear Reviewer 

I have modified the paper according to your instructions.if any questions, please point out, I will modify it again.Thanks 

yours sincerely